# Study on the Properties of Fiber/Matrix Interface and Strain-Hardening Behavior of ECC Containing Municipal Solid Waste Incineration (MSWI) Powder

**DOI:** 10.3390/ma15144905

**Published:** 2022-07-14

**Authors:** Yun Dong, Yongzhen Cheng, Hao Lu

**Affiliations:** 1Faculty of Architecture and Civil Engineering, Huaiyin Institute of Technology, Huai’an 223001, China; dyun@hyit.edu.cn (Y.D.); 230139226@seu.edu.cn (Y.C.); 2College of Water Conservancy and Hydropower Engineering, Hohai University, Xikang Road No.1, Nanjing 210098, China

**Keywords:** MSWI, ECC, pull-out test, strain-hardening index

## Abstract

In this paper, the mechanical properties of micropowder cement mortar and engineered cementitious composites (ECC), using different processing municipal solid waste incineration (MSWI) as a mineral admixture, were investigated. Through the direct ball milling method, ball milling heat treatment method, water washing ball milling method and water washing heat treatment ball milling method, the mechanical properties of MSWI bottom slag-regenerated micropowder cement mortar were tested. Compared with other groups, the flexural strength and compressive strength of the specimen prepared by the MSWI after washing and heating (750 °C, 5 h) were the highest, which reached 82.0% and 81.0% of the reference group, respectively. Based on this treatment, a uniaxial tensile test, three-point bending test and single fiber pull-out test were then carried out to explore the relevant ECC properties containing MSWI. The strain-hardening index PSH of ECC was determined by analyzing the fracture toughness and elastic modulus, fiber/matrix interface chemical bond and friction bond strength of ECC containing MSWI. The results showed that the PSH index of ECC was higher when the treated powder content was 2.2, the w/c ratio was 0.25 and the fiber volume content was 2.0%. This led to higher tensile ductility, which made it easier to achieve stable multi-slit cracking and strain-hardening behavior.

## 1. Introduction

Engineered cementitious composites (ECC) are a special high-performance fiber reinforced cement-based material. ECC materials are designed based on fracture mechanics and microscopic mechanics theories [1,2], and the mechanical properties of ECC are improved by adjusting the fiber, matrix and interface properties of the fiber and matrix. Under the action of bending, ECC materials can show ultra-high deformation capacity and have strain-hardening behavior similar to metal [3].

Since ECC materials were successfully designed and prepared in the 1990s, researchers around the world have carried out in-depth research on ECC materials. Li et al. [4] found that the elastic modulus and diameter of fibers would affect the tensile properties of ECC. When fibers with a large diameter and low elastic modulus were selected, the fibers were not easy to fracture, but these factors were also conducive to improving the mechanical properties of ECC materials. ECC showed obvious strain-hardening characteristics under tensile action when the fiber volume ratio was 2.0%. The parallel and fine crack development showed excellent crack control ability, and the maximum crack width of ECC was only around 60 μm, which effectively improved the durability of the material. Fine aggregate can improve the elastic modulus of ECC, but excess fine aggregate affected the strain hardening performance and ductility of ECC [5]. In the past, Sahmaran et al. [6] found that when fiber was evenly dispersed in the matrix, ECC fracture toughness changed little as the aggregate size increased. Using a uniaxial tensile test, Kim et al. [7] found that the tensile strain of ECC mixed with blast furnace slag reached 3.6% and the ultimate tensile strength reached 5.2 MPa. In addition, fiber length also had an impact on the tensile properties of ECC materials. When a fiber with a length of 12 mm was added, ECC not only maintained reasonable tensile strength, but also had good tensile ductility and toughness.

In order to improve the working performance of ECC paste and improve the dispersion of fibers, so as to effectively control the toughness of the matrix and strain-hardening behavior of ECC materials, a coarse aggregate was not used in ECC materials [8]. As a result, ECC materials had relatively high cement content, leading to greater hydration heat and material shrinkage. To solve this problem, a large amount of fly ash was used in the ECC materials instead of cement. Fly ash can effectively reduce the slip-hardening effect of fibers in the matrix, and the fine particles of fly ash adhered to the surfaces of fibers and isolated fibers from hydration products [9]. Huang et al. [10] studied the preparation of ECC with a large amount of fly ash and proposed a green and environmentally friendly ECC. The early compressive strength of the environmentally friendly ECC was similar to that of ordinary cement concrete, but the strength increased rapidly in the later period and the tensile strain could reach more than 4%. Yang et al. [11] believe that using fly ash to partially replace cement in ECC can increase the friction and adhesion of the fiber/matrix interface, which reduces the fracture toughness and free drying shrinkage of the matrix, and is conducive to enabling ECC to obtain good tensile properties and fine cracks. Emily and Li [12] found that ECC materials still had high tensile properties after six months of exposure in a natural environment.

Bottom residue refers to residues discharged from the end of the incineration bed after municipal solid waste incineration (MSWI), whose mass was found to be close to 80% of the total weight of ash and was the main component of ash [13,14]. The MSWI slag can be used as a partial substitute for cement and concrete aggregate, which has been widely used in many countries, such as the United States, Japan, and other countries [15,16,17]. Alderete [18] tested the fresh and hardened properties of concrete by replacing some Portland cement with ground domestic waste incineration bottom ash. Accordingly, a mix proportion for a concrete mixture with good performance was designed, and the overall sustainability of concrete was quantified according to the preliminary life cycle assessment. In addition, using MSWI bottom slag to treat solidified sludge not only solved the problem of municipal solid waste treatment, but also solved the problem of solidified sludge, which accords with the concept of green environmental protection.

Based on the reasonable improvement of the hydration activity of MSWI bottom slag, the microscopic design parameters of recycled ECC after MSWI bottom slag was used as a cement substitute material were analyzed in this paper. The influence of MSWI slag-regenerated micropowder on the mechanical properties of ECC was discussed, which clarified the mechanical properties and mechanism of MSWI slag-regenerated ECC.

## 2. Materials and Test Methods

### 2.1. Materials

The MSWI slag used in the experiment was taken from Xuyi incineration power plant in Huai’an, Jiangsu Province. The chemical composition of MSWI slag was analyzed by an X-ray fluorescence analyzer, as shown in Table 1. According to the test results in Table 1, SiO_2_, Al_2_O_3_, Fe_2_O_3_, and CaO in MSWI bottom slag accounted for around 70% of the total mass of the bottom slag, which belonged to a typical CaO-SiO_2_-Al_2_O_3_-Fe_2_O_3_ chemical system [19]. The particle frequency distribution curve and cumulative distribution curve of MSWI bottom slag are shown in Figure 1. It is shown that most of the MSWI particle sizes were concentrated in the 4.75–1.18 micron range, accounting for around 66.1% of the total particles. The cumulative contents of 20.0%, 60.0%, and 80.0% of particle size were around 4.75, 2.25, and 0.60 micron, respectively.

The ordinary Portland cement used in this test was PO 42.5-grade Portland cement produced by Conch Cement Co., LTD, Huai’an, China. The chemical composition of the cement is shown in Table 2. Xiamen ISO standard sand was selected for the MSWI cement mortar strength test, and its SiO_2_ content was not less than 98.0% of natural round silica sand. 

MSWI bottom slag-recycled ECC material used refined quartz sand as a coarse aggregate, with a diameter of around 100 μm and particle size of 150 mesh. The fiber used in this paper was polyvinyl alcohol fiber (PVA), whose important properties are shown in Table 3. The water-reducing agent used was commercial CQJ-JSS model polycarboxylic acid high-efficiency water-reducing agent, and its water-reducing rate was more than 20.0%.

### 2.2. Test Method

#### 2.2.1. Strength Test of Mortar Containing MSWI

The MSWI ball mill adopts the xQM-4 planetary ball mill produced by Changsha Tianchuang Powder. The ball mill is shown in Figure 2a. The specific four processing steps are as follows: direct ball milling method (Q/BA), using a planetary ball mill for the direct ball milling test of MSWI raw materials, with a grinding speed and time of 20 r/min and 40 min; ball milling heat treatment (QT/BA), on the basis of the micropowder obtained by direct ball milling, with heat treatment at 750 °C for 5 h; the water washing ball milling method (WQ/BA), in which we soak the MSWI raw materials in water to remove impurities, and then grind according to the above steps; the ball milling method (WTQ/BA), in which the bottom slag after washing and removing impurities is first heat-treated, and then ground according to the same process as in method 1. The processed MSWI bottom slag-regenerated micropowder is shown in Figure 2b.

The chemical composition of the treated powder is shown in Table 4. The main oxides of the treated powder are similar to cement, and the main components belong to the typical CaO-SiO_2_-Al_2_O_3_-Fe_2_O_3_ system. The total amount of Q/BA, WQ/BA, QT/BA, and WTQ/BA oxides accounted for 85.53%, 85.36%, 84.05%, and 84.91%, respectively. However, compared with cement, the content of CaO in the powder was lower, while the content of SiO_2_, Al_2_O_3_, and Fe_2_O_3_ was higher, leading to good pozzolanic activity in the later stage.

The treated MSWI bottom slag micropowder was divided into three specifications of micropowder, namely 80–120 mesh, 120–200 mesh, and less than 200 mesh, and the strength test of MSWI mortar was conducted, respectively. The strength test of mortar containing MSWI was strictly in accordance with GB/T17671-1999 [20] to measure its 3 d, 7 d, 28 d bending strength and compressive strength, and the equipment used in the test was the DKZ-6000 electric bending test machine and an automatic constant pressure compression test machine. Multiple groups of cement mortar specimens of 40 mm × 40 mm × 160 mm were mixed with a constant water–cement ratio of 0.5, and the mixing amount of recycled micropowder from MSWI bottom slag was 0.0%, 10%, 20%, and 30% of the total amount of cement (cement + micropowder), respectively. The specific mix ratio design is shown in Table 5.

#### 2.2.2. Uniaxial Tensile Properties of Regenerated ECC Containing MSWI

The feeding sequence and mixing time of ECC raw materials had an important impact on the working performance. After repeated tests and adjustments, the preparation method of recycled ECC from MSWI bottom slag was as follows. (1) The quality of water was measured, and then the weighed polycarboxylic acid water-reducing agent was added to the water and stirred evenly for subsequent use. (2) The cement, quartz sand, and powder were weighed according to the mixing ratio, and then poured into the mixing pot and stirred for 2 min until the three materials were evenly mixed. (3) The water dissolved by the polycarboxylic acid water-reducing agent was poured into the stirring pot, which was stirred at a low speed for 4 min and then stirred quickly for 2 min. (4) PVA fiber was slowly added into the slurry. It was stirred at low speed for 2 min to fully disperse the fiber, and then it was stirred at high speed for 1 min. All materials were evenly dispersed. After the new mixture worked well, the ECC mixing process of MSWI bottom slag regeneration was completed. 

The specimen size for the ECC uniaxial tensile test of MSWI bottom slag regeneration was 330 mm × 60 mm × 13 mm, as shown in Figure 3. The mixing ratio of micropowder, water–binder ratio, and PVA fiber volume ratio were taken as the main influencing factors, which were divided into 9 groups in total, with 3 dumbbell-shaped specimens in each group. The mixing ratio is shown in Table 6. The test instrument adopted a long electronic universal testing machine with a maximum load of 100 kN and a displacement loading speed of 0.5 mm/min.

#### 2.2.3. Three-Point Bending Test of Regenerated ECC Containing MSWI

The three-point bending test was performed with 350 mm × 76 mm × 38 mm specimens. Since the three-point bending test only produced specimens without a fiber matrix, the matrix mixture ratio was divided into 7 groups. The mixture ratio was consistent with the mixture amount of D1–D7 in Table 6, with the mixture amount of micropowder and water–binder ratio as the changing parameters. The test adopted centralized loading, the test span was 200 mm, and the displacement control loading speed was 0.5 mm/min. The displacement sensor was fixed with a magnetic bearing. The displacement data were measured by a DH3820 static strain test system, and the data collection frequency was set at 2.0 Hz, as shown in Figure 4. 

#### 2.2.4. Single Fiber Pull-Out Test of Regenerated ECC Containing MSWI

The single fiber pull-out test of ECC with MSWI included two processes: specimen making and single fiber pulling out. In the production stage of test specimens, a mold with reserved holes was used. After the fiber passed through the mold, ECC materials were poured into PVA fibers buried below 1.0 mm. After curing to the specified age, ECC materials were cut to obtain specimens that met the test requirements. The single fiber pull-out test of ECC materials was carried out on a small-range pull-out test machine, including the fixation of specimens, the bonding of fibers, and the adjustment of the base to place the fibers in a vertical state, and then the fiber pull-out test was carried out under a certain control displacement. The size of the single fiber pull-out specimen was approximately 10 mm × 5 mm × 1 mm. The single fiber pull-out test was divided into 7 groups, as with the three-point bending test, and the mixing amount of micropowder and water–binder ratio were taken as the changing parameters, which were consistent with the mixing amount of D1–D7 in Table 6. 

## 3. Results and Discussion

### 3.1. Strength Test of Mortar Containing MSWI

The influence of powder content on the compressive strength of cement mortar is shown in Table 7 and Figure 5. The compressive strength of mortar decreased with the increase in the replacement amount of micropowder; that is, the mortar specimen with 10% micropowder had the highest compressive strength, second only to that without micropowder, while the mortar specimen with 30% micropowder had the lowest compressive strength. For constant powder content, WTQ/BA had the highest compressive strength compared with the other three treatments, followed by ball milling QT/BA compressive strength, while direct ball milling and water washing ball milling had lower compressive strength. Among all the components, the compressive strength of the powder with 10% WTQ/BA was the highest, with a value of 59.42 MPa. The influence of the amount of regenerated powder in bottom slag on the flexural strength of mortar was highly consistent with the compressive strength. This may be due to the fact that the heat treatment at 750 °C caused the organic matter in the powder to burn, reducing the adverse impact on the hydration of the cement and leading to the ultimate enhancement of its mechanical properties [21]. The mortar specimen mixed with WTQ/BA had higher flexural strength and compressive strength than the mortar specimen mixed with QT/BA, which was due to the effect of the ball milling sequence and heat treatment on the particle size distribution of the fine powder, resulting in the smaller particle size of WTQ/BA compared to QT/BA and finally showing better mechanical properties. 

The flexural and compressive strength of the powder increased with the increase in the fineness of the powder, as displayed in Table 8 and Figure 6. The compressive strength and flexural strength of WTQ/BA specimens with a fineness of 200 mesh reached the maximum in each group, which were 49.16 MPa and 6.89 MPa, respectively. Finer particles not only can improve the compactness of mortar specimens, but also enhance the activity of MSWI bottom slag to regenerate micropowder. As reported by previous studies [22,23], the chemical composition of MSWI residue-regenerated micropowder changes greatly with particle size, especially SiO_2_ and CaO. The finer the particles are, the lower the SiO_2_ content is, and the higher the CaO content is. 

The influence of curing age on mortar strength is shown in Table 9 and Figure 7. It can be found that with the increase in curing age, the flexural strength and compressive strength of the mortar specimen could be greatly improved, which was caused by the continuous hydration of the material. When the curing age was 3 d, 7 d, and 28 d, the flexural strength of WTQ/BA under different treatment methods remained high, especially when the curing age was 28 d, where the flexural strength of WTQ/BA reached 6.89 MPa. At all curing ages, the flexural strength and compressive strength of all mortar specimens were lower than those of the control specimens, which was due to the higher hydration activity of cement compared with MSWI. 

Taking 70% of the strength value of the standard curing for 28 d of the control mortar as the qualified value of the admixture, it can be found that only the compressive strength and flexural strength of the WTQ/BA mortar specimen met the requirements, which were reduced by 20.28% and 17.29%, respectively, compared with the reference group, while the QT/BA test group occupied second place and only the compressive strength met the requirements. In addition, compared with the control group, the early strength of mortar mixed with micro mortar was relatively low, but the strength increased rapidly in the later period, which was related to the pozzolanic reaction of the MSWI material in the later period [24,25].

### 3.2. Uniaxial Tensile Performance Test with Recycled ECC Containing MSWI

Figure 8 shows the influence of micropowder content on the tensile properties of regeneration ECC containing MSWI bottom slag. When the powder content was 1.2%, the stress–strain curve began to rise rapidly and linearly, and the ECC material was in the elastic rising stage. After reaching the ultimate tensile strength, there was no stable serrated fluctuation, and then it decreased rapidly. The whole stress–strain curve had no obvious strain-hardening phenomenon. The stress–strain curve of the D1 group kept a similar trend when the content of fine powder increased to 1.5%. Meanwhile, the tensile strain increased rapidly and the tensile stress remained stable at the late stage of saw tooth fluctuation when the content of fine powder increased to 2.2%, and the ultimate tensile strain exceeded 2.0%. At this time, the ECC specimen was found in a stable state of multi-crack cracking. The development of microcracks caused the ECC to show a zigzag pattern of rising or falling in the strain-hardening stage. Before the microcrack was formed, the tensile stress of a section was borne by both the matrix and fiber. The development of microcracks disconnected the matrix and released the stress on the matrix, making the tensile stress drop instantly. Then, due to the bridging of the fibers, the tensile stress gradually increased until the next microcrack appeared.

The initial cracking strength and ultimate tensile strength of ECC decreased with the increase in micropowder content. The initial cracking strength and ultimate tensile strength of ECC were 3.07 MPa and 3.63 MPa when the content of micropowder was 1.2%, while the initial cracking strength and ultimate tensile strength of ECC were only around 1.84 MPa and 2.06 MPa when the content of micropowder was 4.0%.

With the increase in micropowder content, the tensile strain-hardening performance of ECC was improved obviously. When the micropowder content was 1.2%, the ultimate tensile strain of ECC was around 1.0%. Meanwhile, when the micropowder content was 4.0%, the ultimate tensile strain of ECC increased to 2.0%. It can be seen that 2.2% was the best mixture. The mixture not only had ductility and tensile properties, but also maintained reasonable tensile strength. However, compared with traditional fly ash ECC, there was a large gap of 5% in the ultimate tensile strain [26]. According to the study of Wang and Li [8], the concentration of metal ions at the interface between fiber and matrix controls the chemical bonding force at the interface, especially the role of aluminum ions and calcium ions, which form a strong bonding layer between the PVA fiber and cement particles [27]. The higher Al_2_O_3_ and CaO content tends to increase the concentrations of A1^3+^ and Ca^2+^ at the fiber/matrix interface [28], leading to higher bonding at the interface. According to the existing literature [26], the Al_2_O_3_ content in fly ash is 22.64~35.99%, and the CaO content is around 1.65~8.55%. In WTQ/BA, the Al_2_O_3_ content is 10.02%, and the CaO content is 19.09%. Although the content of CaO in fine powder is higher than that in fly ash, Ca(OH)_2_ is released during cement hydration to make up for the deficiency of CaO in fly ash. The reason that the tensile strain of the WTQ/BA composite was smaller than that of traditional ECC is that the content of active component Al_2_O_3_ was lacking.

Figure 9 shows the influence of the w/c ratio on the tensile properties of recycled ECC containing MSWI bottom slag. When the w/c ratio was 0.2, the ECC material maintained high tensile strength, but the serrations in the “saw tooth” stage were few and the “saw tooth” width was narrow, indicating that only a few or fine cracks were generated in the specimen, which was consistent with the macroscopic cracking of the material. When the w/c ratio increased to 0.25, the tensile strength of ECC began to increase, and the ultimate tensile strain also showed a rising trend. The tensile strain-hardening property of ECC was significantly improved, and the ultimate tensile strain could be stabilized at more than 2.0%, and the tensile strength was also higher. When the w/c ratio increased to 0.3, there was significant bleeding in the matrix, and although the obtained curves showed significant strain hardening and uniform surface fiber dispersion, the tensile strength decreased significantly. The comprehensive analysis showed that a low w/c ratio tended to cause fiber agglomeration, which was not conducive to the chaotic distribution and strain-hardening characteristics of PVA fibers. The higher w/c ratio led to poor adhesion and increased porosity between substrates, so the w/c ratio should be controlled at 0.25. Under this w/c ratio, the recycled ECC containing MSWI waste residue showed good tensile ability.

When the w/c ratio increased from 0.2 to 0.3, the initial cracking strength of ECC decreased from 2.32 MPa to 1.56 MPa, and the ultimate tensile strength decreased from 2.68 MPa to 1.78 MPa. The high w/c ratio led to the increase in free water in the suspension and the increase in the number of holes in the matrix, leading to a decrease in uniformity and compactness. Once the PVA fiber was pulled out prematurely in the tensile process, the stress–strain curve entered the stress strengthening stage in advance. In addition, the increase in w/c ratio also reduced the chemical bond between the PVA fiber and matrix, and the slip between cracks increased the crack width. If the load continued to increase, the width of the crack increased, thereby replacing the appearance of new cracks. The fibers continued to slip until friction occurred. In contrast, a low w/c ratio contributed to the uniform dispersion of PVA fibers, which maximized the fiber bridging force and thus achieved stable cracks in the matrix.

Figure 10 shows the influence of the fiber volume ratio on the tensile properties of recycled ECC containing MSWI bottom slag. When the fiber volume fraction was 1.5%, the curve rose smoothly to the highest point and then dropped rapidly to the lowest point. There was only one crack in the sample, and there was no strain-hardening phenomenon. When the fiber content was 2.0%, the curve of the sample was “serrated”, indicating that the number of cracks had increased and the strain-hardening phenomenon was obvious. When the fiber volume ratio was 2.5%, the ultimate tensile strain increased continuously. In addition, when the fiber volume ratio increased from 1.5% to 2.5%, the initial cracking strength increased from 1.86 MPa to 2.33 MPa, and the ultimate tensile strength increased from 2.4 MPa to 3.0 MPa. The matrix at both ends of the ECC microcrack was connected by the PVA fiber, and the fiber crossing the crack after the matrix cracking limited the development of the crack. The stress transfer between the fiber and matrix led to the formation of multiple fine cracks in the ECC, improving the overall tensile property of the matrix. In particular, the ultimate tensile strain of ECC at 2.5% PVA volume ratio reached 2.27%, which was more than 200 times that of ordinary concrete.

### 3.3. Three-Point Bending Test of Recycled ECC Containing MSWI Slag

The load–displacement curves of the three-point bending tests with different WTQ/BA and w/c ratios are shown in Figure 11. As the load gradually increased, the curve rose almost linearly to the maximum load. The intersection of the line and the curve near the maximum load was the initial crack load, and then the curve rose briefly until the maximum load occurred, and the load–displacement curve began to show a zigzag shape. At this time, the load of the specimen did not exceed the previous maximum load and remained for a period of time, showing a certain ductility. The load–displacement curve decreased rapidly until the specimen lost its bearing capacity. With the increase in the differential dosage from 1.2% to 4.0%, the obvious “peak and valley” before reaching the maximum load showed a trend of first increasing and then decreasing. The three-point bending specimen of ECC with 2.2% micropowder content exhibited the best strain-hardening characteristics under tension at the lower end; this material had good toughness and ductility at this ratio. According to the three-point bending test of ECC under different w/c ratios, shown in Figure 10b, it could be found that the load–displacement curve had the largest number of peaks and valleys and the best strain-hardening characteristics when the w/c ratio was 0.25.

### 3.4. Single Fiber Pull-Out Test of Recycled of ECC with MSWI Slag

The failure modes of the single fiber pull-out test of ECC can be divided into two types: complete fiber pull-out and fiber pull-out. In the test of recycled ECC containing MSWI residue, the pull-out specimen with PVA fiber embedded at a depth of 1 mm failed as a whole due to fiber pulling out. Theoretically, there should be no fiber residue on the matrix after fiber pulling out, but in practice, there was still a small amount of fiber residue on the matrix. This was because the drawing force exceeded the tensile force of the PVA fiber when the matrix strength was high, leading to the fracture of some fibers in the specimen. In order to better understand the bond between the fiber and the cement paste, the following analysis only considered the fiber being completely pulled out. The whole process of the PVA fiber from initial debonding to final debonding in the ECC single fiber pull-out test is shown in Figure 12. Among them, the curve starts from the initial zero point to the maximum pull-out load point. In this process, the PVA fiber mainly bonded with the matrix through chemical bonding, and this stage was the initial debonding. With the continuous increase in displacement, the load started to decrease from the maximum pulling load point to zero. This stage was the friction stage, which depended on the friction between the PVA fiber and matrix. When the load was zero, it indicated that the PVA fiber had been completely pulled out of the matrix. 

## 4. Strain-Hardening Mechanism Analysis

### 4.1. Matrix Parameters

According to ECC microscopic design theory, ECC matrix parameters include initial crack strength σ_cr_, elastic modulus *E*_m_, and fracture toughness *K*_m_ [6,11,12]. Among them, the elastic modulus and initial crack strength were obtained by the uniaxial tensile test, and the fracture toughness (*K*_m_) was calculated by the three-point bending test of the ECC matrix; the specific formula is as follows:(1)$$K_{m}=\frac{1.5\left(F_{Q}+\frac{mg}{2}\cdot 10^{-2}\right)\cdot 10^{-3}\cdot S\cdot a_{0}^{1/2}}{th^{2}}f\left(\alpha \right)$$(2)$$f(\alpha )=\frac{1.99-\alpha (1-\alpha )\left(2.15-3.93\alpha +2.7\alpha^{2}\right)}{(1+2\alpha ){\left(1-\alpha \right)}^{3/2}}$$
(3)$$\alpha =\frac{a_{0}}{h}$$where *K*_m_ is the fracture toughness, unit/MPa·m^1/2^; *F_Q_* is the peak load obtained by the three-point bending test, in unit/kN; m is the mass of the specimen, unit/kg; *g* is the acceleration of gravity, 9.8 m/s^2^; *S* is the span of the three-point bending specimen, in unit/m; *T* is the width of the three-point bending beam, unit/m; *H* is the thickness of the three-point bending beam, unit/m; *f*(*α*) is the dimensionless shape parameter of a three-point curved beam. 

The relationship between the initial cracking strength, differential dosage, and w/c ratio of the ECC matrix is shown in Figure 13. The initial cracking strength σ_cr_ of the matrix decreased with the increase in micropowder content. When the micropowder content was 1.2%, the initial cracking strength was 3.07 MPa, and when the micropowder content was 4.0%, the fracture strength was reduced to 1.84 MPa, a total decrease ratio of 40%. This was due to the low level of early hydration products in the matrix with higher content of fine powders. With the increase in the w/c ratio, the initial cracking strength of the matrix increased first and then decreased. An overly low or high w/c ratio resulted in uneven fiber distribution and material bleeding, respectively.

The relationship between the fracture toughness, elastic modulus, differential dosage, and w/c ratio of the ECC matrix is shown in Figure 14. With the increase in powder content from 1.2% to 4%, the fracture toughness of the matrix decreased, while the elastic modulus increased first and then decreased. The reason is that the reaction rate of the micropowder is lower than that of cement, which leads to the reduction in the fracture toughness of the ECC matrix. With the increase in the water–binder ratio from 0.2 to 0.3, the matrix fracture toughness first increased and then decreased, and the elastic modulus also showed a similar trend. A large number of holes existed on the surface of the ECC matrix, and the existence of holes also provided channels for the expansion of ECC matrix microcracks, leading to the formation of small microcracks in the ECC matrix more easily. When the w/c ratio increased from 0.25, the elastic modulus decreased significantly, and the decreasing range of the elastic modulus increased. 

Based on the matrix fracture toughness (*K*_m_) and elastic modulus (*E*_m_) obtained above, the fracture toughness (*J*_tip_) at the crack tip can be obtained according to Formula (4) [5,29,30]. The influence of powder content and w/c ratio on J_tip_ is shown in Figure 15.

The *J*_tip_ decreases obviously with the increase in micropowder content, from 32.8 J/m^2^ (micropowder content is 1.2) to 21.5 J/m^2^ (micropowder content is 4.0), decreasing by 34.5%. *J*_tip_ decreases first and then increases with the increase in w/c ratio. The decrease in crack tip fracture toughness *J*_tip_ indicates that ECC requires less energy for the forward propagation of the crack tip, which is conducive to the stable development of the crack.

The *J*_tip_ decreased obviously with the increase in micropowder content, from 32.8 J/m^2^ (micropowder content was 1.2%) to 21.5 J/m^2^ (micropowder content was 4.0%), with a decrease rate of 34.5%. The *J*_tip_ decreased first and then increased with the increase in w/c ratio. The decrease in crack tip fracture toughness J_tip_ indicated that ECC required less energy for the forward propagation of the crack tip, which was conducive to the stable development of the crack.
(4)$$J_{\mathrm{tip}}=\frac{K_{m}^{2}}{E_{m}}$$
where *K*_m_ is the fracture toughness of ECC, unit/MPa·m^1/2^, *E*_m_ is the elastic modulus obtained by the uniaxial tensile test, unit/GPa; *J*_tip_ is the tip fracture toughness of the ECC matrix, in unit J/m^2^.

### 4.2. Fiber/Matrix Interface Parameters

The material properties of ECC were mainly determined by the matrix, fiber, and the interface characteristics between fiber and matrix. Fiber/matrix parameters generally include chemical bond G_d_, interfacial friction *τ*_0_, and fiber slip-hardening factor β. In this paper, chemical bond G_d_ and interfacial friction *τ*_0_ were studied. The relationship between single fiber pulling force and displacement under different slip-hardening conditions of ECC materials is shown in Figure 16. When the debonding stage between the fiber and the matrix is complete, the tension on the fiber suddenly drops from *P*_a_ to *P*_b_. This may be due to the sudden disappearance of G_d_, the chemical bond between the PVA fiber and matrix, resulting in the unstable propagation of cracks and the sudden drop in external tension. Therefore, G_d_ can be calculated from tension *P*_a_ and tension *P*_b_, as shown in Formula (5). In addition, the frictional bond strength *τ*_0_ between the PVA fiber and substrate can be calculated according to Formula (6).

(5)$$G_{d}=\frac{2{\left(P_{a}-P_{b}\right)}^{2}}{\pi^{2}E_{f}d_{f}}$$(6)$$\tau_{0}=\frac{P_{b}}{\pi d_{f}L_{e}}$$where *L_e_* is the embedding depth of the fiber in the matrix, unit/kN; *P_a_* is the maximum debonding force, unit kN; *P_b_* is the debonding force at the end of the debonding stage between the fiber and the matrix, and the unit is kN; *τ*_0_ stands for slip friction stress between fiber and matrix, unit/kN; G_d_ stands for chemical bonding force, unit/kN; *E_f_* stands for the elastic modulus of the fiber, unit/GPa; *D_f_* is the diameter of the fiber, unit/μm;

The influence of different amounts of powder content on chemical bond G_d_ and friction bond τ_0_ is shown in Figure 17. With the increase in powder content from 1.2% to 4.0%, chemical bond G_d_ and friction bond *τ*_0_ showed a decreasing trend. The chemical bond (G_d_) and friction bond (*τ*_0_) were 2.17 J/m^2^ and 1.73 MPa, respectively, when the powder content was 1.2%. When the powder content was 4.0%, the frictional adhesion *τ*_0_ was 1.50 J/m^2^ and 1.l5 MPa, respectively, which decreased by 30.88% and 34.68%, respectively.

The decrease in G_d_ in the chemical bond with high content of micropowder was much larger than the increase in G_d_ with low content of micropowder, and the decreasing trend of the friction bond strength with high content of micropowder became slow. The lower G_d_ in the chemical bond indicates that the PVA fiber was easier to pull out from the interface during the tensile process. With the increase in the w/c ratio from 0.2 to 0.3, the chemical bond G_d_ and τ_0_ decreased by 44.4% and 45.7%, respectively. When the w/c ratio increased, the bond between the PVA fiber and matrix was weak, and a large number of pores were introduced into the matrix. These pores reduced the hydration products wrapped around the PVA fiber, and also diluted the concentrations of some metal ions in the matrix, thus reducing the chemical bond G_d_ and friction bond *τ*_0_.

### 4.3. Strain-Hardening Performance Index PSH

According to the microscopic design theory of ECC, only under energy criterion Formula (7) and strength criterion Formula (8) under the stress–strain curve can stable multi-crack cracking and strain-hardening behavior of ECC be achieved. The strength criterion controls the crack process and the energy criterion determines the crack development mode.

(7)$$J_{\mathrm{tip}}\leq \sigma_{0}\delta_{0}-\int_{0}^{\delta_{0}}\sigma (\delta )d\delta =J_{b}^{’}$$(8)$$\sigma_{0}>\sigma_{\mathrm{cr}}$$where *σ*_0_ is the maximum bridge bonding force of the fiber, unit/MPa; *δ*_0_ is the displacement corresponding to the maximum bridge bonding force of the fiber, unit/mm; *σ*_cr_ is the initial cracking strength of the material, unit/MPa.

However, in practical engineering applications, the dimensional defects of ECC materials and the disorderly distribution of PVA fibers in the matrix lead to the failure of stable strain hardening and multi-crack cracking behaviors, which makes it difficult to accurately evaluate the actual strength and energy criteria. In 1998, Kanda et al. and Li et al. [28] proposed to use the strain-hardening index PSH (=*J*_b_’/*J*_tip_) as an index to evaluate multi-crack saturation—that is, the state in which no new microcracks can be formed under the action of external tensile forces (the multi-crack saturation state). The minimum value of strain-hardening index PSH can be set to 1, indicating that the ECC material is in a completely saturated state with multiple cracks. Generally, in order to ensure the strain-hardening performance of ECC materials, the strain-hardening index PSH should be preferably between 1 and 2. The larger the strain-hardening index PSH value, the more easily the material can achieve stable multi-crack cracking and strain hardening behavior. According to Kan Lili’s latest research [31], saturation cracking can only be achieved when the strain-hardening index PSH (=*J*_b_’/*J*_tip_) of ECC materials is ≥3. Therefore, in order to obtain materials with high tensile strain capacity, it is necessary to improve the PSH index in the design.

The change in the ECC strain-hardening index PSH is shown in Figure 18. In the past, when the powder content was 1.2% (group 1) and 1.5% (group 2), the PSH index was less than 1, which does not meet the strain-hardening energy criterion. It can be seen that the residual energy of group 1 and 2 external forces cannot cause the crack to continue to expand stably. With the increase in external forces, the width of the crack also increases rapidly, ultimately leading to the destruction of ECC materials. Although the strain-hardening strength criterion, namely *σ*_0_/*σ*_cr_ >1, was satisfied, the stable strain-hardening property of ECC could not be guaranteed. When the content of fine powder exceeded 2.2% (group 3, group 4, group 5), the PSH index was greater than 1, which meets the basic condition of strain hardening. When the powder content was 2.2%, all the PSH indexes were greater than 1, and both the energy criterion and strength criterion were satisfied, even in the case of different w/c ratios (groups 3, 6, and 7). When the micropowder content was 2.2%, the water–binder ratio was 0.25, and the fiber volume content was 2.0%; the PSH index (*J*_b_’/*J*_tip_) was 1.71, and the increase in the PSH index was conducive to the strain-hardening behavior of ECC and more saturated and stable multi-crack cracking.

Compared with the other groups, the maximum bridge bonding force *σ*_0_ of the three groups met the strength criterion compared with the initial cracking strength *σ*_cr_ obtained by the uniaxial tensile test. The bridge residual energy *J*_b_’ was also greater than the fracture energy *J*_tip_ of the matrix and met the energy criterion. The maximum PSH index was 1.71, which meets the necessary conditions of saturation strain-hardening performance of ECC materials. When the amount of micropowder was 2.2%, the w/c ratio was 0.25, and the fiber volume ratio was 2.0%, the recycled ECC of MSWI bottom slag showed more stable strain hardening and multi-crack cracking behavior.

## 5. Conclusions

The main conclusions of this paper are as follows:(1)Compared with other treatments, the mortar with WTQ/BA-treated MSWI material passed through a 0.075 mm sieve showed the highest mechanical strength at the Portland cement dosage of 30%.(2)With the increasing content of WTQ/BA, the initial cracking strength and ultimate tensile strength decreased, while the strain-hardening performance improved. When the w/c ratio increased from 0.2 to 0.3, the uniaxial tensile properties of ECC increased first and then decreased.(3)When the content of micropowder was 2.2%, the three-point bending test of ECC showed multiple peaks and valleys with zigzag characteristics, where the mixture not only had ductility and tensile properties, but also maintained reasonable tensile strength.(4)According to the single fiber pull-out test of ECC with WTQ/BA content, the chemical bond (G_d_) and friction bond (*τ*_0_) at the interface decreased with the increase in WTQ/BA content and w/c ratio, and the decrease range was between 30.0% and 46.0%.(5)When the micropowder content was 2.2%, the w/c ratio was 0.25, and the fiber volume ratio was 2.0%, the ECC under the micropowder content of WTQ/BA met both the strength criterion and the energy criterion, and the PSH index was the highest. ECC exhibited higher tensile ductility and could more easily achieve stable multi-crack cracking and strain hardening.

## Figures and Tables

**Figure 1 materials-15-04905-f001:**
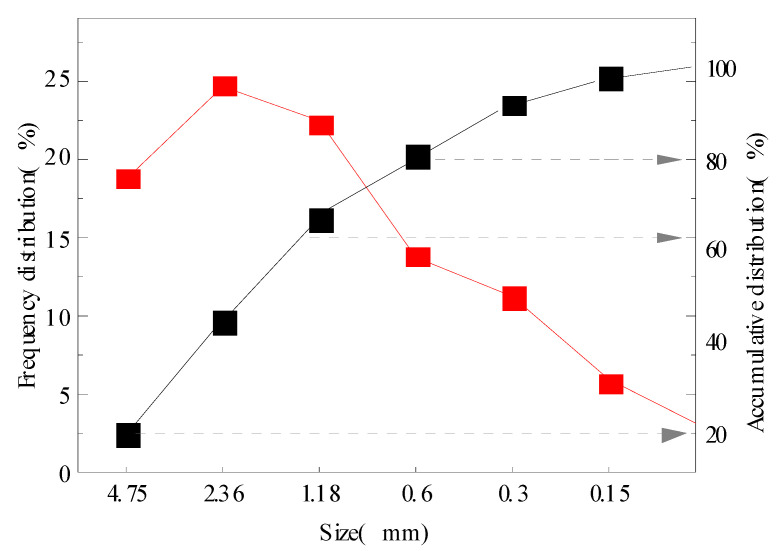
Grading curves of bottom slag particles.

**Figure 2 materials-15-04905-f002:**
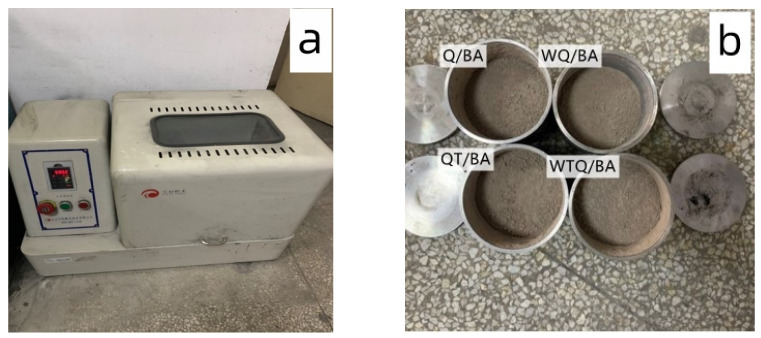
XQM-4 planetary ball mill apparatus (**a**) and the obtained micropowders (**b**).

**Figure 3 materials-15-04905-f003:**
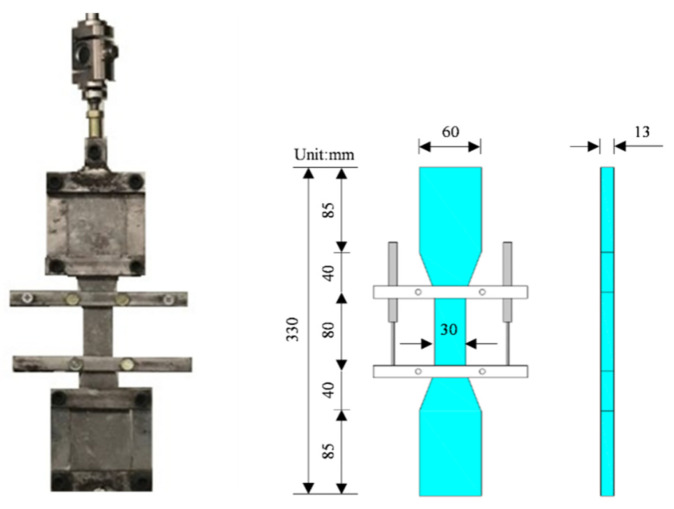
Mold and size of ECC uniaxial tensile test.

**Figure 4 materials-15-04905-f004:**
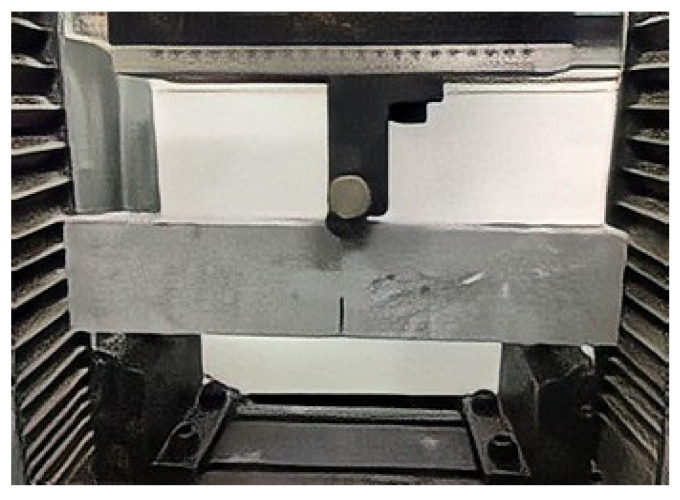
Three-point bending test of regenerated ECC containing MSWI.

**Figure 5 materials-15-04905-f005:**
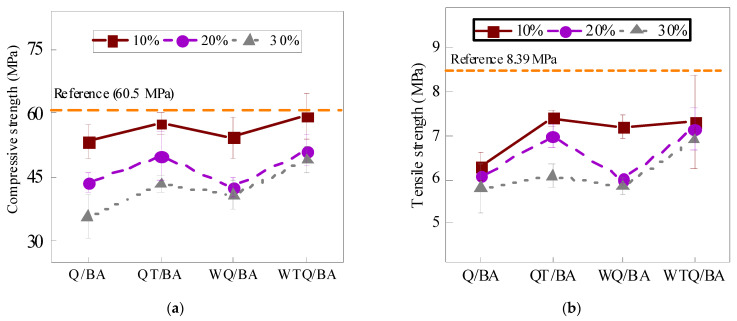
Different contents of Q/BA, QT/BA, WQ/BA, and WTQ/BA in compressive strength (**a**) and flexural strength (**b**).

**Figure 6 materials-15-04905-f006:**
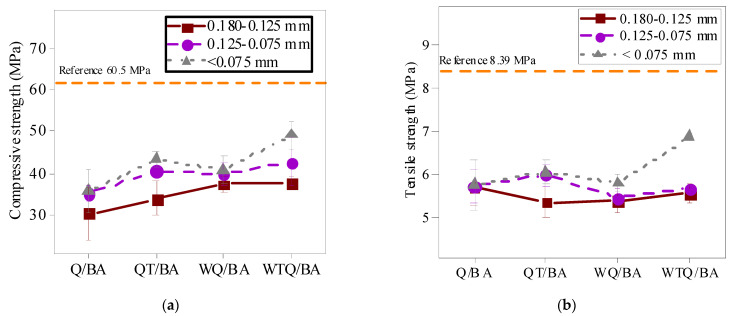
Different particle sizes of Q/BA, QT/BA, WQ/BA, and WTQ/BA in compressive strength (**a**) and flexural strength (**b**).

**Figure 7 materials-15-04905-f007:**
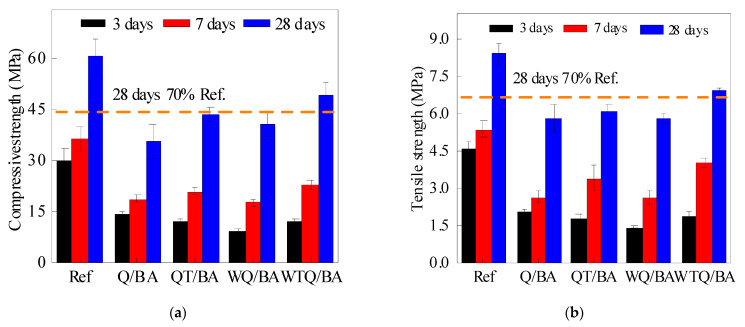
Compressive strength (**a**) and flexural strength (**b**) of MSWI mortars within 3, 7, 28 days of curing of Ref, Q/BA, QT/BA, WQ/BA, and WTQ/BA.

**Figure 8 materials-15-04905-f008:**
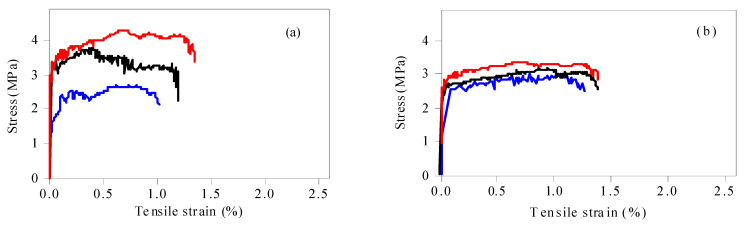
Uniaxial tensile stress–strain curves of ECC specimens with different WTQ/BA content, (**a**–**e**) refer to the 1.2, 1.5, 2.2, 3.0, and 4.0 dosages of MWSI, respectively.

**Figure 9 materials-15-04905-f009:**
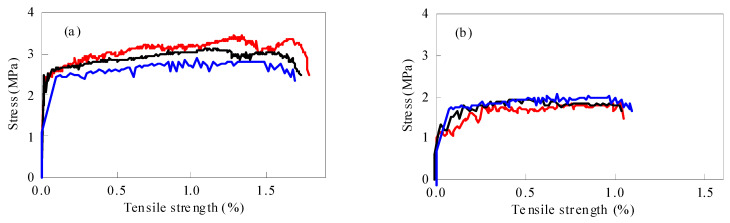
Uniaxial tensile stress–strain curves of ECC specimens with different w/c ratios, (**a**) and (**b**) refer to the 0.20 and 0.30 water-binder ratio, respectively.

**Figure 10 materials-15-04905-f010:**
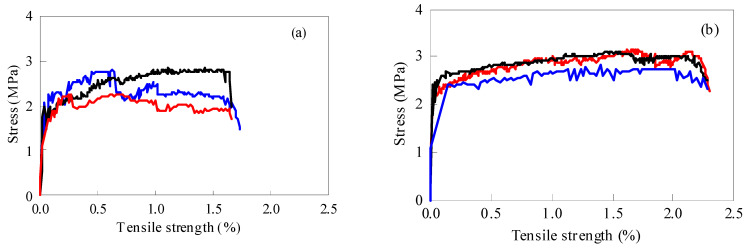
Uniaxial tensile stress–strain curves of ECC specimens with different fiber content, (**a**) and (**b**) refer to the 1.5% and 2.55 fiber dosage, respectively.

**Figure 11 materials-15-04905-f011:**
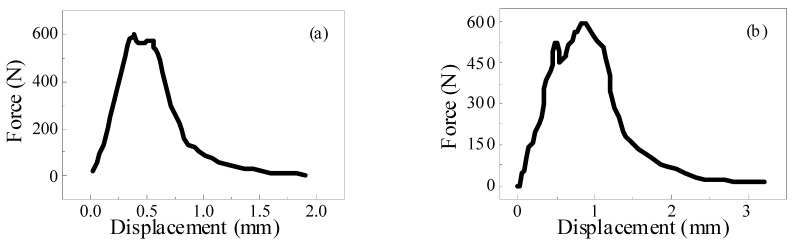
Load–displacement curves of matrix specimens with different MSWI dosage and water-binder ratios, (**a**–**e**), refer to the 1.2, 1.5, 2.2, and 3.0 MSWI dosage; (**f**,**g**) refer to 0.2 and 3.0 water-binder ratio.

**Figure 12 materials-15-04905-f012:**
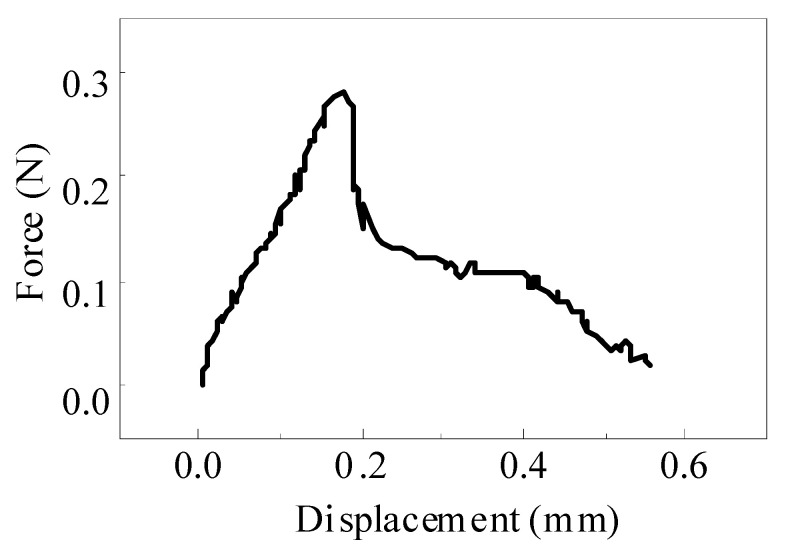
Typical load–displacement curves of ECC single fiber pull-out test.

**Figure 13 materials-15-04905-f013:**
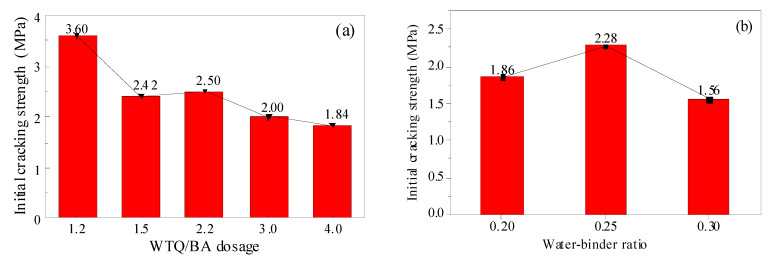
Initial cracking strength of ECC with different WTQ/BA dosages (**a**) and water–binder ratios (**b**).

**Figure 14 materials-15-04905-f014:**
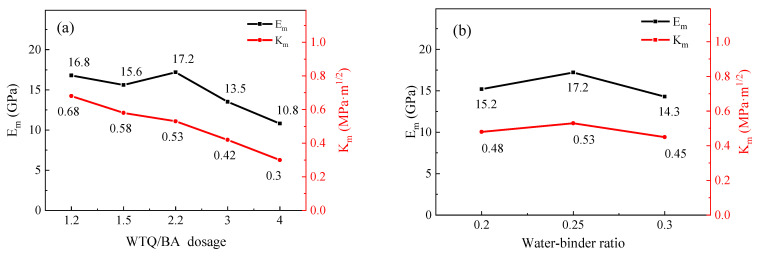
Fracture toughness and elastic modulus of ECC with different WTQ/BA dosages (**a**) and water–binder ratios (**b**).

**Figure 15 materials-15-04905-f015:**
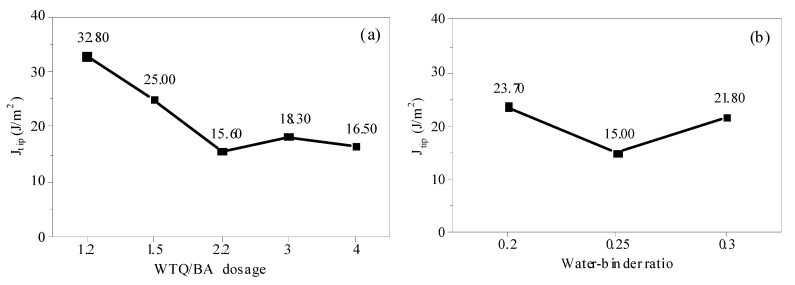
Fracture energy of ECC with different WTQ/BA dosages (**a**) and water–binder ratios (**b**).

**Figure 16 materials-15-04905-f016:**
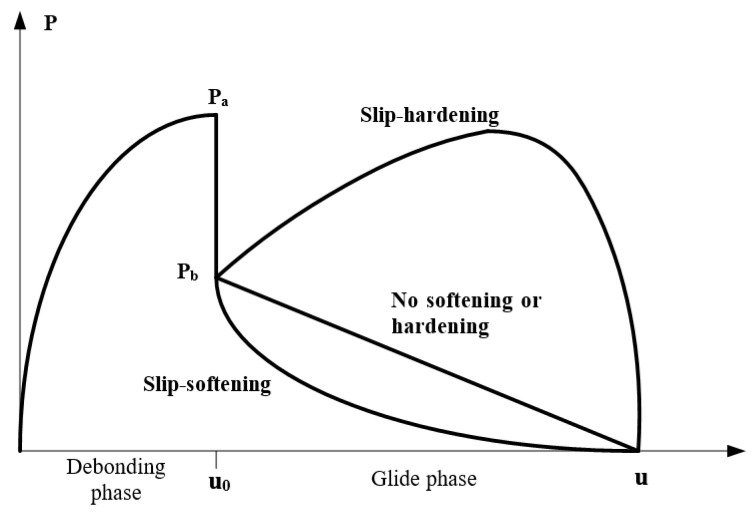
Relationship between pulling force and displacement of single fiber under different slip conditions.

**Figure 17 materials-15-04905-f017:**
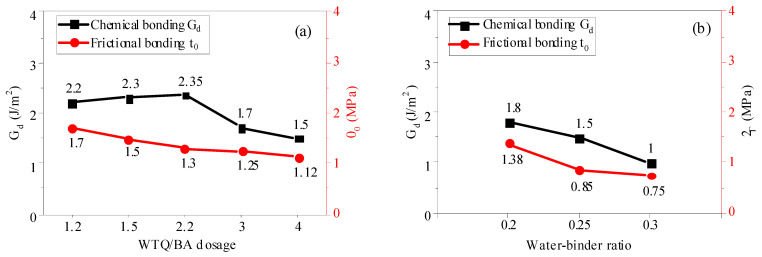
Chemical bonding G_d_ and frictional bonding *τ*_0_ of ECC with different WTQ/BA dosages (**a**) and water–binder ratios (**b**).

**Figure 18 materials-15-04905-f018:**
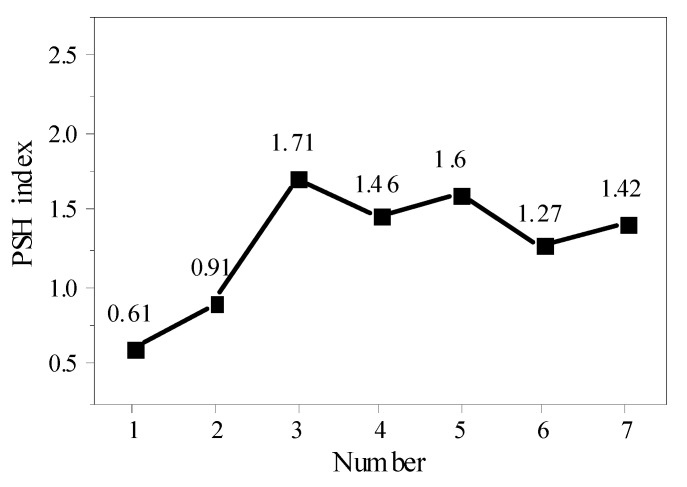
PSH indicators of different group numbers.

**Table 1 materials-15-04905-t001:** Chemical composition of MSWI bottom slag.

Composition	SiO_2_	CaO	Al_2_O_3_	Na_2_O	Fe_2_O_3_	P_2_O_5_	K_2_O	MgO	TiO_2_	Others
Content (%)	48.41	14.78	11.99	3.25	5.40	1.86	1.42	1.78	0.76	10.35

**Table 2 materials-15-04905-t002:** Chemical composition of cement (%).

Composition	SiO_2_	Fe_2_O_3_	Al_2_O_3_	MgO	CaO	SO_3_	K_2_O	Na_2_O	LOI
Content	21.06	3.94	5.47	1.75	62.30	2.64	0.23	0.22	1.61

**Table 3 materials-15-04905-t003:** Main parameters of PVA fiber.

Fiber	Diameter	Density	Length	Ductility	Young’s Modulus	Tensile Strength
	(μm)	(g/cm^3^)	(mm)	(%)	(GPa)	(MPa)
PVA	39.0	1.3	12.0	7.0	42.0	3000

**Table 4 materials-15-04905-t004:** OPC, Q/BA, WQ/BA, QT/BA, and WTQ/BA chemical compositions (Unit: wt.%).

Compositions	CaO	SiO_2_	Fe_2_O_3_	Al_2_O_3_	MgO	K_2_O	Na_2_O	ZnO	CuO	P_2_O_5_	TiO_2_
OPC	56.95	21.92	3.21	8.16	1.66	0.72	0.00	0.23	0.05	0.89	0.25
Q/BA	19.63	48.23	8.18	9.49	2.19	2.39	3.10	0.35	0.12	2.83	0.83
WQ/BA	18.59	49.84	7.39	9.53	2.35	2.33	3.84	0.31	0.13	2.55	0.91
QT/BA	21.32	45.86	7.53	9.34	2.63	2.23	3.54	0.37	0.18	3.40	0.81
WTQ/BA	19.09	49.00	6.80	10.02	2.47	2.41	3.68	0.33	0.16	2.90	0.90

**Table 5 materials-15-04905-t005:** Mix ratio of MSWI mortar test.

Number	MSWI (g)	Cement (g)	Sand (g)	Water (g)
C1	0	450	1350	225
C2	45	405	1350	225
C3	90	360	1350	225
C4	135	315	1350	225

**Table 6 materials-15-04905-t006:** Matrix mix ratio of ECC materials.

Number	Cement	Sand–Binder Ratio	MSWI	Water–Binder Ratio	Water Reducer	Fiber
D1	1	0.36	1.2	0.25	0.02	2.0%
D2	1	0.36	1.5	0.25	0.02	2.0%
D3	1	0.36	2.2	0.25	0.02	2.0%
D4	1	0.36	3.0	0.25	0.02	2.0%
D5	1	0.36	4.0	0.25	0.02	2.0%
D6	1	0.36	2.2	0.2	0.02	2.0%
D7	1	0.36	2.2	0.3	0.02	2.0%
D8	1	0.36	2.2	0.25	0.02	1.5%
D9	1	0.36	2.2	0.25	0.02	2.5%

Note: Fiber dosage is volume ratio, the others are mass ratio.

**Table 7 materials-15-04905-t007:** Mechanical strength for mortar with different MSWI dosages.

Processing	Curing Period (d)	Size (Mesh)	Dosage (%)	FS (MPa)	CS (MPa)
Reference	28		0	8.39	60.5
Q/BA		<200	10	6.29	53.38
		<200	20	6.07	43.78
		<200	30	5.77	38.53
QT/BA		<200	10	7.83	59.36
		<200	20	6.97	9.88
		<200	30	6.02	34.00
WQ/BA		<200	10	7.20	56.26
		<200	20	6.08	42.72
		<200	30	5.49	40.83
WTQ/BA		<200	10	7.31	59.42
		<200	20	7.13	51.28
		<200	30	6.89	49.16

**Table 8 materials-15-04905-t008:** Mechanical strength for mortar with different MSWI particle size dosages.

Processing	Curing Period (d)	Size (Mesh)	Dosage (%)	FS (MPa)	CS (MPa)
Q/BA	28	80–120	30	6.07	30.38
		120–200		5.79	35.23
		<200		5.77	35.73
QT/BA		80–120		6.07	43.38
		120–200		5.39	40.68
		<200		6.00	35.40
WQ/BA		80–120		5.40	37.40
		120–200		5.82	39.90
		<200		5.49	40.83
WTQ/BA		80–120		5.58	37.90
		120–200		5.69	42.67
		<200		6.89	49.16

**Table 9 materials-15-04905-t009:** Mechanical strength for mortar with different curing ages.

Processing	Curing Period (d)	Size (Mesh)	Dosage (%)	FS (MPa)	CS (MPa)
Reference	3			4.86	30.25
	7			5.38	36.37
	28			8.33	60.50
Q/BA	3	<200	30	2.06	14.12
	7	<200		2.65	19.05
	28	<200		5.77	38.52
QT/BA	3	<200		1.76	12.03
	7	<200		3.42	21.13
	28	<200		6.00	34.00
WQ/BA	3	<200		1.43	9.57
	7	<200		2.65	17.85
	28	<200		5.49	40.83
WTQ/BA	3	<200		1.91	12.18
	7	<200		4.06	23.10
	28	<200		6.89	48.23

## Data Availability

Data sharing is not applicable.

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
