# Peer review of "Study on the Properties of Fiber/Matrix Interface and Strain-Hardening Behavior of ECC Containing Municipal Solid Waste Incineration (MSWI) Powder"

_materials, 2022, doi:10.3390/ma15144905_

Round 1

Reviewer 1 Report

Generraly the paper is intersting but same part must be changed. Specialy writing of value is uncorected.

In the Table 1 the sum is not 100%,

The figure 15 must be rebilt. It is not clear

Too many references are from China or in Chinese language

see attched file

Reviewer 2 Report

This paper studied the replacement of fly ash in ECC with MSWI. The paper was well written and quite clear. Though I have few comments as below:

1. MSWI - please clarify if this is MSWI fly ash or another material.

2. Section 3.1 - please add references when the author described mechanisms such as pozzolanic even though this is a common phenomena. 

Reviewer 3 Report

The paper aims to investigate the mechanical properties of Engineered Cementitious Composites (ECC) after replacing the fly ash with municipal solid waste incineration MSWI. However, major corrections must be done before publishing:

1)    The abstract is blurry and the results must present well based on the methods that used in this paper.

2)    The introduction must improved and references related to using  MSWI must add to reflect the novelty of this study:

“Effective and sustainable use of municipal solid waste incineration bottom ash in concrete regarding strength and durability”

“Municipal solid waste incineration ash-incorporated concrete: one step towards environmental justice”

“A Review on Cementitious Materials Including Municipal Solid Waste Incineration Bottom Ash (MSWI-BA) as Aggregates”

“The Use of Municipal Solid Waste Incineration Ash in Various Building Materials: A Belgian Point of View”

3)    In section 2.1 , this statement “According to the test results in Table 1, SiO2, Al2O3, Fe2O3and CaO in MSWI bottom slag account for about 70% of the total mass of the bottom slag, which belonged to a typical CaO-SiO2-Al2O3-Fe2O3 chemical system” what is the typical chemical system, please refer to the reference?

4)    Please explain more about Figure 1

5)    Please clarify the reasons of using refined quartz sand as coarse aggregate and the fiber polyvinyl alcohol fiber (PVA) based on previous studies.

6)    Table 1 should corrected the zero is missing in most of percentages like “.25, .86 etc.”

7)    Based on the abstract, the MSWI was used to replace the fly ash, however, the Table 5 did not refer any mixes contain fly ash in order to compare and identify the improvement of using MSWI instead of fly ash. Other mixes contain fly ash must add.

8)    In section 3.1 this statement must corrected to” The influence of curing age on mortar strength is shown in Figure 6” Figure 9 is present the uniaxial tensile stress

9)    The discussions of the all results are weak and must supply by references.

10)           The author indicated that WTQ/BA with 10% dosage of micro powder is the highest compressive strength and flexural strength at 28 days. However, Table 7 indicated that the flexural strength of QT/BA with 10% micropowder recorded the highest value. Please clarify?

11)           The titles of Figures 4, 6 and Table 8 are burry. Please modify. Abbreviations are not recommended in the titles.

12)           Extensive English language is recommended

13)           The titles of figures and tables must modify and formatting.

Reviewer 4 Report

Chapter 5 is large and needs to be shortened, leaving only novelty and results. Move everything else to the previous section.

The information presented in fig. 12(c) and fig. 13(c) is unconvincing since it is built on only three components. This point should be strengthened.

Shown in Fig. 4 identical information should be displayed either in the text or in the caption. The same applies to Fig. 5, 6.

References are made in the text of the article to fig. 6 and in fig. 9 which is located much further than fig. 7 and 8. The text should be revised.

You need a sequence of references to the figure. There are no references in the text to Fig. 5.

What does the number (1) in subsection 2.2.2 mean. because the number (2) in the text of such a subsection is not

The dimensions of the specimens for uniaxial tension are indicated simultaneously in text 3. It would be better if fig. 3 should be removed completely and replaced with a 3-point bend test circuit.

In figure 1, it is not clear how on what scale the graphs relate to the y-axis. It is necessary to draw direction arrows on the appropriate scale on them.

Round 2

Reviewer 3 Report

The authors have addressed all the suggested comments and the paper can be further published